

# Clustering assessment in weighted networks

Argimiro Arratia and Martí Renedo Mirambell

Department of Computer Sciences, Polytechnical University of Catalonia, Barcelona, Catalonia, Spain

## ABSTRACT

We provide a systematic approach to validate the results of clustering methods on weighted networks, in particular for the cases where the existence of a community structure is unknown. Our validation of clustering comprises a set of criteria for assessing their significance and stability. To test for cluster significance, we introduce a set of community scoring functions adapted to weighted networks, and systematically compare their values to those of a suitable null model. For this we propose a switching model to produce randomized graphs with weighted edges while maintaining the degree distribution constant. To test for cluster stability, we introduce a non parametric bootstrap method combined with similarity metrics derived from information theory and combinatorics. In order to assess the effectiveness of our clustering quality evaluation methods, we test them on synthetically generated weighted networks with a ground truth community structure of varying strength based on the stochastic block model construction. When applying the proposed methods to these synthetic ground truth networks' clusters, as well as to other weighted networks with known community structure, these correctly identify the best performing algorithms, which suggests their adequacy for cases where the clustering structure is not known. We test our clustering validation methods on a varied collection of well known clustering algorithms applied to the synthetically generated networks and to several real world weighted networks. All our clustering validation methods are implemented in R, and will be released in the upcoming package *clustAnalytics*.

## INTRODUCTION

Clustering of networks is a popular research field, and a wide variety of algorithms have been proposed over the years. However, determining how meaningful the results are can often be difficult, as well as choosing which algorithm better suits a particular data set. This paper focuses specifically on weighted networks (that is, those in which the connections between nodes have an assigned numerical value representing some property of the data), and we propose novel methods to validate the community partitions of these networks obtained by any given clustering algorithm. In particular, our clustering validation methods focus on two of the most important aspects of cluster assessment: the significance and the stability of the resulting clusters.

Corresponding author
Argimiro Arratia,
argimiro@cs.upc.edu

We consider clusters produced by a clustering algorithm to be *significant* if there are strong connections within each cluster, and weaker connections (or fewer edges) between different clusters. This notion can be quantified and formalized by applying several community scoring functions (also known as quality functions in *Fortunato (2010)*), that gauge either the intra-cluster or inter-cluster density. Then, it can be determined that the partition of a network into clusters is significant if it obtains better scores than those for a comparable network with uniformly distributed edges.

On the other hand, *stability* measures how much a clustering remains unchanged under small perturbations of the network. In the case of weighted networks, these could include the addition and removal of vertices, as well as the perturbation of edge weights. This is consistent with the idea that meaningful clusters should capture an inherent structure in the data and not be overly sensitive to small and/or local variations, or the particularities of the clustering algorithm.

Our goal is to provide a systematic approach to perform these two important clustering validation criteria, which can be used when the underlying structure of a network is unknown, because in this case different algorithms might produce completely different results, and it is not trivial to determine which ones are more adequate, if any at all.

To assess the significance of communities structure, in general weighted networks, we provide a collection of community scoring functions that measure some topological characteristics of the ground-truth communities as defined by *Yang & Leskovec (2015)* for unweighted networks. Most of these topological characteristics focus on the relation between the external and internal connectivity of clusters, density of edges and degree distributions. Our scoring functions are proper extensions of those in *Yang & Leskovec (2015)* to weighted networks. A separate case is the clustering coefficient, a popular scoring function in the analysis of unweighted networks. We examined several existing definitions for the weighted case, being most relevant to us the descriptions given by *Barrat et al. (2004)*; *Saramäki et al. (2007)*, and *McAssey & Bijma (2015)*, and found the latter to be the most versatile (for instance, it can be used in complete graphs where all the information is given by the values of the edges, such as those generated from correlation networks). The clustering coefficient of *McAssey & Bijma (2015)* is defined in terms of an integral, and we provide an efficient way of computing it. Then, to evaluate the significance (in a statistical sense) of the scores produced by any scoring function, we compare them against null models with similar properties but without any expectations of a community structure. For this we propose an extension to weighted graphs of the switching model (*Milo et al., 2003*) which produces random graphs by rewiring edges while maintaining the vertex degree sequence. The idea is that a significant community in any given network should present much better scores than those of the randomly generated ones.

As for the stability of clusters, it has been studied more widely for algorithms that work on Euclidean data (as opposed to networks, weighted or not). For instance, *Von Luxburg (2010)* uses both resampling and adding noise to generate perturbed versions of the data. *Hennig (2007)* introduces bootstrap resampling (with and without perturbation) to evaluate cluster stability. Also for Euclidean data, *Vendramin, Campello & Hruschka (2010)* introduce a systematic approach for cluster evaluation that combines cluster quality criteria

with similarity and dissimilarity metrics between partitions, and searches for correlations between them. Our approach consists of a bootstrap technique with perturbations adapted to clustering on networks, that resembles what Hennig does for Euclidean data. That is, the set of vertices is resampled multiple times, and the clustering algorithms are applied to the resulting induced networks. In this case, the perturbations are applied to the edge weights after resampling the vertices, but the standard bootstrap method without perturbation can be used on all networks, weighted or not.

To compare how the clusters of the resampled networks differ from the originals, we use three measures. The adjusted Rand index (*Hubert & Arabie, 1985*) is a similarity measure that counts the rate of pairs of vertices that are in agreement on both partitions, corrected for chance. Additionally, we use measures derived from information theory to compare partitions such as the recently introduced Reduced Mutual Information (*Newman, Cantwell & Young, 2020*), which corrects some of the issues with the original mutual information or its normalized version (*Danon et al., 2005*). For example, giving maximal scores when one of the partitions is trivial, which in our case would mean that failed algorithms that split most of the network into single vertex clusters would be considered very stable. Other attempts at providing adjusted versions of the mutual information include *Dom (2002)*; *Vinh, Epps & Bailey (2010)* and *Zhang (2015)*. The other information theory measure we employ for the sake of comparison and control is the Variation of Information (VI) (*Meilă, 2007*). The VI is a distance measure (as opposed to a similarity measure, like the Rand index and mutual information) that actually satisfies the properties of a proper metric.

We apply these clustering quality evaluation methods on several real world weighted networks together with a varied collection of well known clustering algorithms. Additionally, we also test them on synthetic weighted networks based on the stochastic block model construction (*Holland, Laskey & Leinhardt, 1983*; *Wang & Wong, 1987*), which allows us to have predefined clusters whose strength can be adjusted through a parameter, and which we can compare to the results of the algorithms for their evaluation.

Our main contributions are the following: a switching model for randomizing weighted networks while maintaining the degree distribution, and its use together with the scoring functions we adapted to the weighted case, to provide a general approach to the validation of significance of clustering results. An implementation of a bootstrap method with perturbation adapted to weighted networks, to test for stability. A model to generate benchmark weighted networks based on the stochastic block model, which we use to test our methodology for stability and significance of clusters. Additionally we contribute with an R package *clustAnalytics*, which contains all the functions and methods for cluster analysis that are explained in this paper.

The remainder of this paper is organized as follows. 'Materials and Methods' contains the details of our methods for assessing clustering significance and stability, as well as a description of the clustering algorithms we will put to test and the datasets. 'Discussion' presents the discussion of our experiments. 'Conclusions' presents our conclusions. Finally, we put in an Appendix ("Appendix") the technical details on the time complexity of our algorithms. Our experiments results figures are reported separately in the Table S1 file, which the reader can consult conveniently.

## MATERIALS AND METHODS

To determine if the partition of a graph into communities given by a clustering algorithm provides significant results, we use the scoring functions defined in 'Community scoring functions'. Our method consists in evaluating these functions on clusters produced by a given algorithm on both the original graph and on multiple samples of randomized graphs generated from the original graph (see 'Randomized graph'). Then, for each function we see how the score of the original graph clusters compares to the scores of the randomized graph clusters, as we can define a relative score (score of the original graph over the mean of the scores of the random graphs). We also observe the percentile rank of the original score of the graph in the distribution of scores from the randomized graphs. Depending on the nature of the scoring function, a significant cluster structure will be associated with percentile ranks either close to 1 (for scores in which higher is better) or 0 (when lower is better).

For testing cluster stability, we implement a bootstrap resampling on the set of vertices of the network, plus the addition of a perturbation to the weights of the edges in the induced graph. The details of this methodology are described in 'Bootstrap with perturbation'. The Variation of Information (VI), Reduced Mutual Information (RMI) and Adjusted Rand Index (ARI) introduced in 'Materials and Methods' are used as similarity measures. Then, the bootstrap statistics are the values of these similarity measures comparing the resampled bootstrap graphs to the original one.

On our experiments we evaluate the results of clustering on a selection of networks with different community structure ('Data') with several well-known clustering algorithms ("Clustering algorithms"). Additionally, we also test the algorithms on synthetic graphs with a preset community structure constructed using stochastic block models ("Synthetic ground truth models"). By varying one of the parameters of the model ($\lambda$), we generate networks that range from being mostly uniform (that is, with no community structure) to having very strong communities. This allows us to see how our evaluation methods respond in a controlled environment where the existence or not of strong clusters in the network is known.

### Community scoring functions

Here we will provide functions which will evaluate the division of networks into clusters, specifically when the edges have weights. Using the scoring functions for communities in unweighted networks given in *Yang & Leskovec (2015)* as a reference, we propose generalizations of most of them to the weighted case.

[1] For every variable or function defined over the unweighted graph, will use a " $\sim$ " to denote its weighted counterpart

### *Basic definitions*

Let $G(V, E)$ be an undirected graph of order $n = |V|$ and size $m = |E|$. In the case of a weighted graph[1] $\tilde{G}(V, \tilde{E})$, we will denote $\tilde{m} = \sum_{e \in \tilde{E}} w(e)$ the sum of all edge weights. Given $S \subset G$ a subset of vertices of the graph, we have $n_S = |S|$, $m_S = |\{(u, v) \in E : u \in S, v \in S\}|$, and in the weighted case $\tilde{m}_S = \sum_{(u,v) \in \tilde{E} : u, v \in S} w((u, v))$. We use $w_{uv}$ instead of $w((u, v))$. Note that if we treat an unweighted graph as a weighted graph with weights 0 and 1 (1 if two vertices are connected by an edge, 0 otherwise), then $m = \tilde{m}$ and $m_S = \tilde{m}_S$ for all $S \subset V$.

**Table 1  Community scoring functions $f(S)$ for weighted and unweighted networks.**

| | Unweighted $f(S)$ | weighted $f(S)$ |
|---|---|---|
| ↑ Internal density | $\dfrac{m_S}{n_S(n_S-1)/2}$ | $\dfrac{\tilde{m}_S}{n_S(n_S-1)/2}$ |
| ↑ Edges Inside | $m_S$ | $\tilde{m}_S$ |
| ↑ Average Degree | $\dfrac{2m_S}{n_S}$ | $\dfrac{2\tilde{m}_S}{n_S}$ |
| ↓ Expansion | $\dfrac{c_s}{n_s}$ | $\dfrac{\tilde{c}_s}{n_s}$ |
| ↓ Cut Ratio | $\dfrac{c_s}{n_s(n-n_s)}$ | $\dfrac{\tilde{c}_s}{n_s(n-n_s)}$ |
| ↓ Conductance | $\dfrac{c_s}{2m_s+c_s}$ | $\dfrac{\tilde{c}_s}{2\tilde{m}_s+\tilde{c}_s}$ |
| ↓ Normalized Cut | $\dfrac{c_s}{2m_s+c_s}+\dfrac{c_s}{2(m-m_s)+c_s}$ | $\dfrac{\tilde{c}_s}{2\tilde{m}_s+\tilde{c}_s}+\dfrac{\tilde{c}_s}{2(\tilde{m}-\tilde{m}_s)+\tilde{c}_s}$ |
| ↓ Maximum ODF | $\max_{u\in S}\dfrac{|\{(u,v)\in E:v\notin S\}|}{d(u)}$ | $\max_{u\in S}\dfrac{\sum_{v\notin S}w_{uv}}{\tilde{d}(u)}$ |
| ↓ Average ODF | $\dfrac{1}{n_s}\sum_{u\in S}\dfrac{|\{(u,v)\in E:v\notin S\}|}{d(u)}$ | $\dfrac{1}{n_s}\sum_{u\in S}\dfrac{\sum_{v\notin S}w_{uv}}{\tilde{d}(u)}$ |

Associated to $G$ there is its adjacency matrix $A(G)=(A_{ij})_{1\leq i,j\leq n}$ where $A_{ij}=1$ if $(i,j)\in E$, $0$ otherwise. We insist that $A(G)$ only take binary values 0 or 1 to indicate existence of edges, even in the case of weighted graphs. For the weights we will always use the weight function $w((i,j))=w_{ij}$.

The following definitions will also be needed later on:

- $c_S=|\{(u,v)\in E:u\in S, v\notin S\}|$ is the number of edges connecting $S$ to the rest of the graph.
- $\tilde{c}_S=\sum_{(u,v)\in E:u\in S, v\notin S}w_{uv}$ is the natural extension of $c_S$ to weighted graphs; the sum of weights of all edges connecting $S$ to $G\setminus S$.
- $\tilde{d}(u)=\sum_{v\neq u}w_{uv}$ is the natural extension of the vertex degree $d(u)$ to weighted graphs; the sum of weights of edges incident to $u$.
- $d_S(u)=|\{v\in S:(u,v)\in E\}|$ and $\tilde{d}_S(u)=\sum_{v\in S}w_{uv}$ are the (unweighted and weighted, respectively) degrees[2] restricted to the subgraph $S$.
- $d_m$ and $\tilde{d}_m$ are the median values of $d(u)$, $u\in V$[3].

The left column in Table 1 shows the community scoring functions for unweighted networks defined in *Yang & Leskovec (2015)*. These functions characterize some of the properties that are expected in networks with a strong community structure, with more ties between nodes in the same community than connecting them to the exterior. There are scoring functions based on internal connectivity (internal density, edges inside, average degree), external connectivity (expansion, cut ratio) or a combination of both (conductance, normalized cut, and maximum and average out degree fractions). Uparrow (respectively, downarrow) indicates the higher (resp., lower) the scoring function value the stronger the clustering.

On the right column we propose generalizations to the scoring functions which are suitable for weighted graphs while most closely resembling their unweighted counterparts. Note that for graphs which only have weights 0 and 1 (essentially unweighted graphs) each pair of functions is equivalent (any definition that did not satisfy this would not be a generalization at all).

[2] We assume the weight function $w_{uv}$ is defined for every pair of vertices $u$, $v$ of the weighted graph, with $w_{uv}=0$ if there is no edge between them.

[3] To prevent confusion between the function $d_S(\cdot)$ and the median value (which only depends on $G$) $d_m$ we will always refer to subgraphs of $G$ with uppercase letters.

- **Internal Density, Edges Inside, Average Degree:** These definitions are easily and naturally extended by replacing the number of edges by the sum of their weights.
- **Expansion:** Average number of edges connected to the outside of the community, per node. For weighted graphs, average sum of edges connected to the outside, per node.
- **Cut Ratio:** Fraction of edges leaving the cluster, over all possible edges. The proposed generalization is reasonable because edge weights are upper bounded by 1 and therefore relate easily to the unweighted case. In more general weigthed networks, however, this could take values well over 1 while lacking many "potential" edges (as edges with higher weights would distort the measure). In general bounded networks (with bound other than 1) it would be reasonable to divide the result by the bound, which would result in the function taking values between 0 and 1 (0 with all possible edges being 0 and 1 when all possible edges reached the bound).
- **Conductance and Normalized Cut:** Again, these definitions are easily extended using the methods described above.
- **Maximum and Average Out Degree Fraction:** Maximum and average fractions of edges leaving the cluster over the degree of the node. Again, in the weighted case the number of edges is replaced by the sum of edge weights.

Some of the introduced functions (internal density, edges inside, average degree, clustering coefficient) take higher values the stronger the clusterings are, while the others (expansion, cut ratio, conductance, normalized cut, out degree fraction) do the opposite.

### *Clustering coefficient*

Another possible scoring function for communities is the clustering coefficient or transitivity: the fraction of closed triplets over the number of connected triplets of vertices. A high internal clustering coefficient (computed on the graph induced by the vertices of a community) matches the intuition of a well connected and cohesive community inside a network, but its generalization to weighted networks is not trivial.

There have been several attempts to come up with a definition of the clustering coefficient for weighted networks. One is proposed in *Barrat et al. (2004)* and is given by $c_i = \frac{1}{\tilde{d}(i)(d(i)-1)} \sum_{j,h} \frac{w_{ij}+w_{ih}}{2} A_{ij} A_{jh} A_{ih}$. Note that this gives a local(*i.e.* defined for each vertex) clustering coefficient.

While this may work well on some weighted networks, in the case of complete networks (e.g., such as those built from correlation of time series as in *Renedo & Arratia (2016)*, we obtain

$$
\begin{aligned}
c_i &= \frac{1}{\tilde{d}(i)(d(i)-1)} \sum_{j,h} \frac{w_{ij}+w_{ih}}{2} \\
&= \frac{\sum_{jh} w_{ij} + \sum_{jh} w_{ih}}{\tilde{d}(i)(n-2)\cdot 2} = \frac{(n-2)\sum_j w_{ij} + (n-2)\sum_h w_{ih}}{\tilde{d}(i)(n-2)\cdot 2} \\
&= \frac{(n-2)\tilde{d}(i) + (n-2)\tilde{d}(i)}{\tilde{d}(i)(n-2)\cdot 2} = 1,
\end{aligned}
\tag{1}
$$

which does not give any information about the network.

An alternative was proposed in *McAssey & Bijma (2015)* with complete weighted networks in mind(with weights in the interval $[0,1]$), which makes it more adequate for our case.

- For $t \in [0,1]$ let $A_t$ be the adjacency matrix with elements $A_{ij}^t = 1$ if $w_{ij} \geq t$ and 0 otherwise.
- Let $C_t$ the clustering coefficient of the graph defined by $A_t$.
- The resulting weighted clustering coefficient is defined as

$$\tilde{C} = \int_0^1 C_t \, dt \qquad (2)$$

For networks where the weights are either not bounded or bounded into a different interval than $[0,1]$, the most natural approach is to simply take

$$\tilde{C} = \frac{1}{\bar{w}} \int_0^{\bar{w}} C_t \, dt, \qquad (3)$$

where $\bar{w}$ can be either the upper bound or, in the case of networks with no natural bound, the maximum edge weight. The computation of this integral, which can be expressed as a sum of the values of $C_t$ (a finite amount) is detailed in the "Weighted clustering coefficient".

It is a desirable property that the output of scoring functions remain invariant under uniform scaling, that is, if we multiply all edge weights by a constant $\phi > 0$, as the community structure of the network would be the same. This holds for all of the measures of the third group, which combine the notions of internal and external connectivity.

This means that these scores will be less biased in favour of networks with high overall weight (for the internal connectivity based scores) or low overall weight (for the external connectivity ones). It is particularly interesting for networks with weights that are not naturally upper bounded by one, and facilitates comparisons between networks with completely different weight distributions. When we compare each network's scores to those of a randomized counterpart generated by the switching model, though, the total weight is kept constant, so even scores without this property could still give valuable information.

Let $\tilde{G}_\phi(V, \tilde{E}_\phi)$ be the weigthed graph obtained by multiplying all edge weights in $\tilde{E}$ by real positive number $\phi$. In this case, $n_{S_\phi} = n_S$, $m_{S_\phi} = \phi m_S$, $c_{S_\phi} = \phi c_S$, $d_{S_\phi}(u) = \phi d_S(u)$. This means that the internal density, edges inside, average degree, expansion and cut ratio behave linearly (with respect to their edge weights). Conductance, normalized cut and maximum and average out degree fractions, on the other hand, remain constant under these transformations. Since the notion of community structure is generally considered in relation to the rest of the network (a subset of vertices belong to the same community because they are more connected among themselves than to vertices outside of the community), it seems reasonable to consider that the same partitions on two graphs whose weights are the same up to a multiplicative positive constant factor have the same scores. This makes the scores in the third group, the only ones for which this property holds, more adequate in principle.

For the chosen definition of clustering coefficient this property also holds, as all terms in the integral in equation (3) behave linearly (the proof is immediate with a change of variables), and that linear factor cancels out.

### *Modularity*

As for the modularity (*Newman, 2006b*), it is defined as:

$$Q = \frac{1}{2\tilde{m}} \sum_{ij} \left[ w_{ij} - \frac{\tilde{d}(i)\tilde{d}(j)}{2\tilde{m}} \right] \delta(c_i, c_j). \tag{4}$$

Then, by multiplying the edges by a constant $\phi > 0$, we get the graph $\tilde{G}_\phi(V, \tilde{E}_\phi)$ of modularity:

$$
\begin{aligned}
Q_\phi &= \frac{1}{2\tilde{m}\phi} \sum_{ij} \left[ \phi w_{ij} - \frac{\phi^2 \tilde{d}(i)\tilde{d}(j)}{2\tilde{m}\phi} \right] \delta(c_i, c_j) \\
&= \frac{1}{2\tilde{m}\phi} \phi \sum_{ij} \left[ w_{ij} - \frac{\tilde{d}(i)\tilde{d}(j)}{2\tilde{m}} \right] \delta(c_i, c_j) = Q,
\end{aligned}
\tag{5}
$$

which means that modularity is also invariant under uniform scaling.

## Randomized graph

The algorithm proposed here to generate a random graph which will serve as a null model is a modification of the switching algorithm described in *Milo et al. (2003)*, *Rao, Jana & Bandyopadhyay, (1996)*. It produces a graph with the same weighted degree sequence as the original, but otherwise as independent from it as possible. Each step of this algorithm involves randomly selecting two edges $AC$ and $BD$ and replacing them with the new edges $AD$ and $BC$ (provided they did not exist already). This leaves the degrees of each vertex $A, B, C$ and $D$ unchanged while shuffling the edges of the graph.

One way to adapt this algorithm to our weighted graphs (more specifically, complete weighted graphs, with weights in $[0, 1]$) is, given vertices $A$, $B$, $C$ and $D$, transfer a certain weight $\bar{w}$ from $w_{AC}$ to $w_{AD}$, and from $w_{BD}$ to $w_{BC}$[4]. We will select only sets of vertices $\{A, B, C, D\}$ such that $w_{AC} > w_{AD}$ and $w_{BD} > w_{BC}$, that is, we will be transferring weight from "heavy" edges to "weak" edges. For any value of $\bar{w}$, the weighted degree of the vertices remains constant, but if it is not chosen carefully there could be undesirable side effects.

### *Selection of $\bar{w}$*

We distinguish between two types of weighted networks: those with an upper bound on the possible values of their edge weights given by the nature of the data (usually 1, such as in the Forex correlation network –see 'Data' below), and those without (such as social networks where edge weights count the number of interactions between nodes). Networks with negative weights have not been studied here, so 0 will be a lower bound in all cases.

However, in the case of networks which are upper and lower bounded, this results in a very large number of edge weights attaining the bounds, which might be undesirable (particularly networks like the Forex network, in which very few edges, if any, have weights 0 or 1) and give new randomized graphs that look nothing like the original data.

[4]Recall $w_{ij}$ refers to the weight of the edge between vertices $i$ and $j$

The value of $\bar{w}$ that most closely translates the essence of the switching method for unweighted graphs would perhaps be the maximum that still keeps all edges within their set bounds. This method seems particularly suited to sparse graphs with no upper bound, because it eliminates (by reducing its weight to zero) at least an edge per iteration. Other methods without this property could dramatically increase the edge density of the graph, constantly adding edges by transferring weight to them, while rarely removing them.

However, in the case of very dense graphs such as the Forex correlation network (or any other graph similarly constructed from a correlation measure), this method results in a large number of edge weights attaining the bounds (and in the case of the lower bound 0, removing the edge), which can reduce this density dramatically.

As an alternative, to produce a new set of edges with a similar distribution to those of the original network, we can impose the sample variance (i.e., $\frac{1}{n-1}\sum_{i,j=1}^{n}(w_{ij}-m)^2$, where $m$ is the mean) to remain constant after applying the transformation, and find the appropriate value of $\bar{w}$. The variance remains constant if and only if the following equality holds:

$$
\begin{aligned}
& (w_{AC}-m)^2+(w_{BD}-m)^2+(w_{AD}-m)^2+(w_{BC}-m)^2 \\
= \ & (w_{AC}-\bar{w}-m)^2+(w_{BD}-\bar{w}-m)^2+ \\
& (w_{AD}+\bar{w}-m)^2+(w_{BC}+\bar{w}-m)^2 \\
\Longleftrightarrow \ & 2\bar{w}^2+\bar{w}(-w_{AC}-w_{BD}+w_{AD}+w_{BC})=0.
\end{aligned}
\tag{6}
$$

The solutions to this equation are $\bar{w}=0$ (which is trivial and corresponds to not applying any transformation to the edge weights) and $\bar{w}=\frac{w_{AC}+w_{BD}-w_{AD}-w_{BC}}{2}$.

While this alternative can result in some weights falling outside of the bounds, in the networks we studied it is very rare, so it is enough to discard these few steps to obtain the desired results.

Figure 1 shows how the graph size decreases as the algorithm iterates with the maximum weight method, which also produces a dramatic increase in the variance. The constant variance method on the other hand does not remove any edges and the the size stays constant (as well as the variance, which is constant by definition, so the their corresponding lines coincide at 1).

However, applying the constant variance method on networks that are sparsely connected (such as most reasonably big social networks) results in a big increase in the graph size, to the point of actually becoming complete weighted graphs (see Fig. 2). Meanwhile, the maximum weight method does not significantly alter the size of the graph.

Therefore, we will use the constant variance method only for very densely connected networks, such as correlation networks, which are in fact complete weighted graphs. For sparse networks, the maximum weight method will be the preferred choice.

Note that if all edge weights are either 0 or 1, in both cases this algorithm is equivalent to the original switching algorithm for discrete graphs, as in every step the transferred weight will be one if the switch can be made without creating double edges, or zero otherwise (which corresponds to the case in which the switch cannot be made).

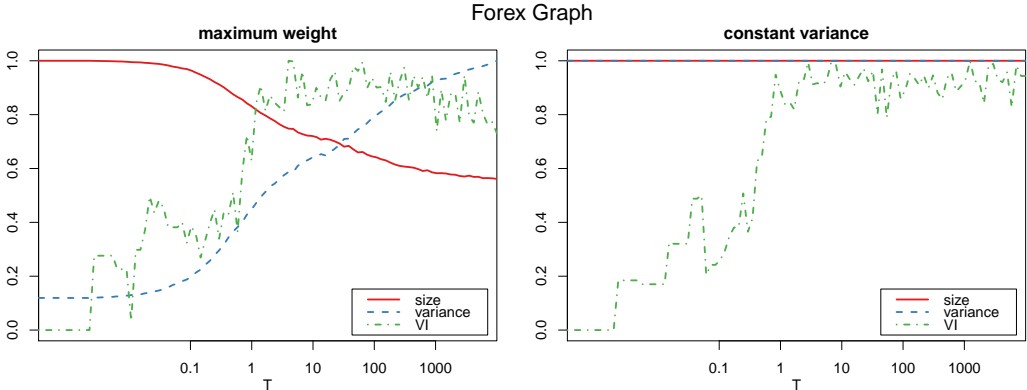

**Figure 1** **Normalized size, variance and variation of information for the Louvain clustering after applying the proposed algorithm on the Forex graph.** The horizontal axis is on logarithmic scale.

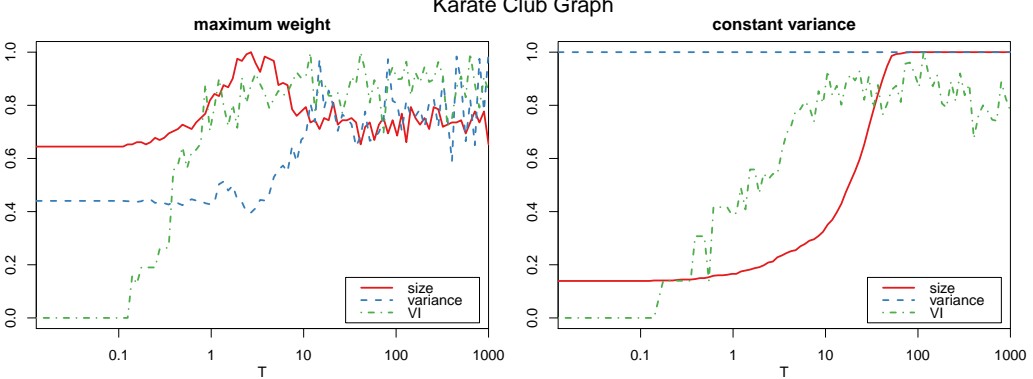

**Figure 2** **Normalized size, variance and variation of information for the Louvain clustering after applying the proposed algorithm on the karate club graph.** The horizontal axis is on logarithmic scale.

### Number of iterations

To determine the number $Tm$ of iterations for the algorithm to sufficiently "shuffle" the network (where $m$ is the size of the graph, and $T$ a parameter we select), we study the *variation of information* (*Meilă, 2007*) of the resulting clustering (in this case using the Louvain algorithm, though other clustering algorithms could be used instead) with respect to the initial one. (In "Clustering similarity measures" we discuss variation of information, and other clustering similarity metrics that we use in this work, and in "Clustering algorithms" we detail all the clustering algorithms that we put to test.) Figs. 1 and 2 show a plateau where the variation of information stops increasing after around $T = 1$ (which corresponds to one iteration per edge of the initial graph). This is consistent with the results for the original algorithm in *Milo et al. (2003)* for unweighted graphs, and we can also select $T = 100$ as a value that is by far high enough to obtain a sufficiently mixed graph.

## Bootstrap with perturbation

Non-parametric bootstrap, with and without perturbation or ''jittering'', has been used to study the stability of clusters of euclidean data sets (*Hennig, 2007*). For graphs, bootstrap resampling can be done on the set of vertices, and then build the resampled graph with the edges that the original graph induces on them (i.e. two resampled vertices will be joined by an edge if and only if they were adjacent in the original graph, with the same weight in the case of weighted graphs). As for adding noise to avoid duplicate elements, it can be added to the edge weights. We suggest generating that noise from a normal distribution truncated to stay within the bounds of the edge weights of each graph (which means it can be truncated on one or both sides depending on the graph).

Then, to deal with copies of the same vertex on the resampled graph, it seems necessary to add heavy edges between them to reflect the idea that a vertex and its copy should be similar and well connected between each other. Not doing so would incentivize the clustering methods to separate them in different clusters, because they generally try to separate poorly connected vertices. We can distinguish two cases:

- Graphs with edge weights built from correlations or other similar graphs which by their nature have a specific upper bound on the edge weights (usually 1): We assign the value of the upper bound to the edge weight. After applying the perturbation, this will result in a weight which will be close to that upper bound.
- Other weighted graphs, where no particular upper bound to the edge weights is known: To assign these edges very high weights (to reflect the similarity that duplicate vertices should have in the resampled network) within the context of the network, one option is to sample values from the highest weights (*e.g.* the top 5%) of the original edge set.

## Synthetic ground truth models

Another way of comparing and assessing the fit of a clustering algorithm is to compare it to a ground truth community structure if there is one, which is seldom known in reality. Alternatively one can synthetically generate a graph with a ground truth community structure. This will allow us to verify that the results of the algorithm match the expected outcome. For the particular case of time series correlation networks one can generate the time series using a suitable model that imposes a community structure with respect to correlations, such as the Vector Autoregressive (VAR) model construction in *Arratia & Cabaña (2013)*, and then compute the values of the edges accordingly.

A common benchmark for clustering algorithm evaluation is the family of graphs with a pre-determined community structure generated by the *l-partition model* (*Condon & Karp, 2001*; *Girvan & Newman, 2002*; *Fortunato, 2010*). It is essentially a block-based extension of the Erdös-Renyi model, with $l$ blocks of $g$ vertices, and with probabilities $p_{in}$ and $p_{out}$ of having edges within the same block and between different blocks respectively.

A more general approach is the stochastic block model (SBM) (*Holland, Laskey & Leinhardt, 1983*; *Wang & Wong, 1987*), which uses a probability matrix $P$ (which has to be symmetric in the undirected case) to determine probabilities of edges between blocks. $P_{ij}$ will be the probability of having an edge between any given pair of vertices belonging to blocks $i$ and $j$ respectively. Then, having higher values in the diagonal than in the rest of

the matrix will produce strongly connected communities. Note that subgraph induced by each community is in itself an Erdös-Renyi graph (with $p = P_{ii}$ for the community $i$). This model also allows having blocks of different sizes. While this model can itself be used for community detection by trying to fit it to any given graph (*Lee & Wilkinson, 2019*), here we will simply use it as a tool to generate graphs of a predetermined community structure.

To obtain a weighted SBM (WSBM) graph, we propose a variation of the model which produces multigraphs, which can then be easily converted into weighted graphs by setting all edge weights as their corresponding edge count. In this case, probability matrix of the original SBM will be treated as the matrix of expectations between edges of each pair of blocks. Then, we simply add edges one by one with the appropriate probability (the same at each step) that will allow each weight expectation to match its defined value. By definition, the probability of the edge added at step $k$ to join vertices $i$ and $j$ is given by

$$P(e_k = (i,j)) = \frac{E_{ij}}{\#\text{steps}}, \tag{7}$$

where $E_{ij}$ is the expected number of edges between them given by the expectation matrix. The sum of these probabilities for all vertices must add up to one, which gives

$$\#\text{steps} = \frac{1}{2} \sum (|C_i||C_j|E_{i,j}). \tag{8}$$

Note that the $\frac{1}{2}$ factor is added because we are using undirected graphs, and we do not want to count edges $(i,j)$ and $(j,i)$ twice.

This process produces a binomially distributed weight for each edge, though these distributions are not independent, so independently sampling each edge weight from the appropriate binomial distribution would not be equivalent.

We will use a graph sampled from this model with block sizes $(40, 25, 25, 10)$, with a parametrized expectation matrix:

$$\begin{pmatrix} 0.03\lambda & 0.01 & 0.01 & 0.03 \\ 0.01 & 0.02\lambda & 0.05 & 0.02 \\ 0.01 & 0.05 & 0.02\lambda & 0.01 \\ 0.03 & 0.02 & 0.01 & 0.03\lambda \end{pmatrix}. \tag{9}$$

With $\lambda = 1$ the network will be quite uniform, but as it increases, the high values in the diagonal compared to the rest of the matrix will result in a very strong community structure, which should be detected by the clustering algorithms.

There are other possible extensions of the stochastic block model to weighted networks such as (*Aicher, Jacobs & Clauset, 2014*), which can have edges sampled from any exponential family distribution. While our approach produces Bernoulli distributed edges (which can be approximated by a Poisson distribution in most cases), the edge distributions obtained in *Aicher, Jacobs & Clauset (2014)* are not independent from each other, so the results are not exactly equivalent. For instance, in our case the total network weight is fixed and will not vary between samples.

**Table 2  Contingency table of partitions $\mathscr{P}$ and $\mathscr{P}'$, with labelings $r$ and $s$.**

|  | $\mathscr{P}'_1$ | $\mathscr{P}'_2$ | ... | $\mathscr{P}'_S$ | sum |
|---|---|---|---|---|---|
| $\mathscr{P}_1$ | $c_{11}$ | $c_{12}$ | ... | $c_{1S}$ | $a_1$ |
| $\mathscr{P}_2$ | $c_{21}$ | $c_{22}$ | ... | $c_{2S}$ | $a_2$ |
| ... |  |  |  | ... |  |
| $\mathscr{P}_R$ | $c_{R1}$ | $c_{R2}$ | ... | $c_{RS}$ | $a_r$ |
| Sum | $b_1$ | $b_2$ | ... | $b_S$ | $n = \sum c_{ij}$ |

## Clustering similarity measures

To compare and measure how similar two clusterings of the same network are, we will use two measures based on information theory, the Variation of Information (VI) and the Reduced Mutual Information (RMI), and another more classical measure, the (adjusted) Rand index which relates to the accuracy. All of these measures are constructed upon the contingency table of the labeling, which is summarized in Table 2, and the terms are explained below.

Consider a set of $n$ elements and two labelings or partitions, one labeled by integers $r = 1, \dots, R$ and the other labeled by integers $s = 1, \dots, S$, let's say $\mathscr{P} = \{\mathscr{P}_1, \dots, \mathscr{P}_R\}$ and $\mathscr{P}' = \{\mathscr{P}'_1, \dots, \mathscr{P}'_S\}$. Define $a_r$ as the number of elements with label $r$ in the first partition, $b_s$ the number of elements with label $s$ in the second partition, and $c_{rs}$ be the number of elements with label $r$ in the first partition and label $s$ in the second. Formally,

$$a_r = |\mathscr{P}_r| = \sum_{s=1}^{S} c_{rs} \tag{10}$$

$$b_s = |\mathscr{P}'_s| = \sum_{r=1}^{R} c_{rs} \tag{11}$$

$$c_{rs} = |\mathscr{P}_r \cap \mathscr{P}'_s| \tag{12}$$

Define the probability $P(r)$ (respectively, $P(s)$) of an object chosen uniformly at random has label $r$ (resp. $s$), and the probability $P(r, s)$ that it has both labels $r$ and $s$, that is

$$P(r) = \frac{a_r}{n}, \quad P(s) = \frac{b_s}{n}, \quad P(r, s) = \frac{c_{rs}}{n} \tag{13}$$

### Variation of information

The variation of information between two clusterings, a criterion introduced in *Meilă (2007)*, is defined as follows.

**Definition 2.1** *The entropy of a partition $\mathscr{P} = \{\mathscr{P}_1, \dots, \mathscr{P}_R\}$ of a set is given by:*

$$\mathscr{H}(r) = -\sum_{r=1}^{R} P(r) \log(P(r)), \tag{14}$$

**Definition 2.2** *The mutual information is defined as:*

$$I(r;s) = \sum_{r=1}^{R} \sum_{s=1}^{S} P(r,s) \log\left(\frac{P(r,s)}{P(r)P(s)}\right) \tag{15}$$

**Definition 2.3** *The variation of information of partitions $\mathscr{P}$ and $\mathscr{P}'$ is given by:*

$$VI(r;s) = \mathscr{H}(r) + \mathscr{H}(s) - 2I(r;s) \tag{16}$$

Intuitively, the mutual information measures how much knowing the membership of an element of the set in partition $\mathscr{P}$ reduces the uncertainty of its membership in $\mathscr{P}'$. This is consistent with the fact that the mutual information is bounded between zero and the individual partition entropies

$$0 \le I(r;s) \le \min\{\mathscr{H}(r), \mathscr{H}(s)\}, \tag{17}$$

and the right side equality holds if and only if one of the partitions is a refinement of the other.

Consequently, the variation of information will be 0 if and only if the partitions are equal (up to permutations of indices of the parts), and will get bigger the more the partitions differ. It also satisfies the triangle inequality, so it is a metric in the space of partitions of any given set.

### Reduced mutual information

The mutual information (*Cover & Thomas, 1991*), often in its normalized form is one of the most widely used measures to compare graph partitions in cluster analysis. More recently (*Newman, Cantwell & Young, 2020*) proposed the *Reduced Mutual Information* (RMI), an improved version which corrects the high mutual information values given to quite dissimilar partitions in some cases. For instance, if one of the partition is the trivial one splitting the network into $n$ clusters of one element each, the standard mutual information will always take the maximal value (1, in the case of the normalized mutual information), even if the other is completely different. More generally, any partitions will always have maximal mutual information with all of their filtrations. This is crucial when comparing clustering algorithms, as some algorithms will output trivial partitions into single-element clusters when they fail to find a clustering structure. Therefore, it would not be possible to reliably measure the stability of these clustering methods with the standard mutual information.

Given $r$ and $s$ two labelings of a set of $n$ elements, the Reduced Mutual Information is defined as:

$$\text{RMI}(r;s) = I(r;s) - \frac{1}{n}\log\Omega(a,b) \tag{18}$$

where $\Omega(a,b)$ is an integer equal to the number of $R \times S$ non-negative integer matrices with row sums $a = \{a_r\}$ and column sums $b = \{b_s\}$. Details on how to compute or approximate $\Omega(a,b)$ are given in the "Methods for counting contingency tables".

The Reduced Mutual Information can also be defined in a normalized form (NRMI), in the same way the standard mutual information is, by dividing it by the average of the values of the reduced mutual information of labelings $a$ and $b$ with themselves:

$$\mathrm{NRMI}(r;s) = \frac{\mathrm{RMI}(r;s)}{\frac{1}{2}[\mathrm{RMI}(r;r) + \mathrm{RMI}(s;s)]}$$

$$= \frac{I(r;s) - \frac{1}{n}\log\Omega(a,b)}{\frac{1}{2}[H(r) + H(s) - \frac{1}{n}(\log\Omega(a,a) + \log\Omega(b,b))]} \tag{19}$$

We will use this normalized form to be able to compare more easily the results of networks with different number of nodes, as well as to compare them to other similarity measures.

### *Rand index*

The Rand Index (RI) and the different measures derived from it (*Hubert & Arabie, 1985*) are based on the idea of counting pairs of elements that are classified similarly and dissimilarly across the two partitions $\mathscr{P}$ and $\mathscr{P}'$. There are four types of pairs of elements:

- **type I**: elements are in the same class both in $\mathscr{P}$ and $\mathscr{P}'$
- **type II**: elements are in different classes both in $\mathscr{P}$ and $\mathscr{P}'$
- **type III**: elements are in different classes in $\mathscr{P}$ and in the same class in $\mathscr{P}'$.
- **type IV**: elements are in the same class in $\mathscr{P}$ and different classes in $\mathscr{P}'$.

Then, similar partitions would have many pairs of elements of types I and II (agreements) and few of type III and IV (disagreements). The Rand index is defined as the ratio of agreements over the total number of pairs of elements.

Using the terms of the contingency table (Table 2), the Rand index is given by

$$\mathrm{RI}(r,s) = \binom{n}{2} + 2\sum_r s\binom{n_{rs}}{2} - [\sum_{r=1}^{R}\binom{a_r}{2} + \sum_{s=1}^{S}\binom{b_s}{2}] \tag{20}$$

An adjusted form of the Rand index (*Hubert & Arabie, 1985*) introduces a correction to account for all the pairings that match on both partitions because of random chance. The Adjusted Rand Index (ARI) is defined as:

$$\mathrm{ARI}(r;s) = \frac{\mathrm{Index} - \mathrm{Expected\ Index}}{\mathrm{Maximum\ Index} - \mathrm{Expected\ Index}}, \tag{21}$$

which in terms of the contingency table (Table 2) can be expressed as:

$$\mathrm{ARI}(r;s) = \frac{\sum_{rs}\binom{c_{rs}}{2} - [\sum_r\binom{a_r}{2}\sum_s\binom{b_s}{2}]/\binom{n}{2}}{\frac{1}{2}[\sum_r\binom{a_r}{2} + \sum_s\binom{b_s}{2}] - [\sum_r\binom{a_r}{2}\sum_s\binom{b_s}{2}]/\binom{n}{2}} \tag{22}$$

## Clustering algorithms

We have selected five well known state-of-the-art clustering algorithms based on different approaches, and all suitable for weighted graphs. They will be applied to all of the networks to then evaluate the results:

1. Louvain method (*Blondel et al., 2008*), a multi-level greedy algorithm for modularity optimization. We use the original algorithm, without the resolution parameter (*i.e.* with resolution $\gamma = 1$).

2. Leading eigenvector method (*Newman, 2006a*), based on spectral optimization of modularity.

3. Label propagation (*Raghavan, Albert & Kumara, 2007*), a fast algorithm in which nodes are iteratively assigned to the communities most frequent in their neighbors.

4. Walktrap (*Pons & Latapy, 2005*), based on random walks.

5. Spin-glass (*Reichardt & Bornholdt, 2006*), tries to find communities in graphs via a spin-glass model and simulated annealing.

In any application, the choice of the clustering algorithm will be hugely dependant on the characteristics of the dataset, as well as its size. The methods proposed here, though, can be applied to evaluate any combination of weighted graph and clustering algorithm.

## Data

- **Zachary's karate club:** Social network of a university karate club (*Zachary, 1977*). The vertices are its 34 members, and the edge weights are the number of interactions between each pair of them. In this case, we have a "ground truth" clustering, which corresponds to the split of the club after a conflict, resulting into two clusters.

- **Forex network:** Network built from correlations between time series of exchange rate returns (*Renedo & Arratia, 2016*). It was built from the 13 most traded currencies and with data of January 2009. It is a complete graph of 78 edges (corresponding to pairs of currencies) and has edge weights bounded between 0 and 1.

- **News on Corporations network:** In this network, a list of relevant companies are the nodes, while the weighted edges between them are set by the amount of times they have appeared together in news stories over a certain period of time (in this instance, on 2019-03-13). It has 899 nodes and 13,469 edges.

- **Social network:** A Facebook-like social network for students from the University of California, Irvine (*Opsahl & Panzarasa, 2009*). It has 1899 nodes (students) and 20,296 edges, weighted by the number of characters of the messages sent between users.

- **Enron emails:** a network composed of email communications among Enron employees (*Klimt & Yang, 2004*). The version of the dataset used here is available in the *igraphdata* R package (*Csardi, 2015*), and consists of a multigraph with 184 vertices (users) and 125,409 edges, corresponding to emails between users. We convert it to a an undirected weighted graph by using as weights for the edges the number of edges in the multigraph (*i.e.* the number of emails between the corresponding users).

## Software

All the methods proposed here are implemented in R (*R Core Team, 2020*), and will be released in an upcoming package. This includes all of the significance functions and the adaptations to the existing boostrap methods to make them work on weighted graphs. All the code interacts with igraph objects (*Csardi & Nepusz, 2006*) for easy testing and manipulation of the graphs, as well as allowing the use of already implemented clustering

methods and other existing functions for exploring graphs. The more computationally intensive parts such as the switching model have been written in C++ for better efficiency, and are called from R through Rcpp (*Eddelbuettel & François, 2011*; *Eddelbuettel, 2013*). Our code is available online: https://github.com/martirm/cluster_assessment.

## DISCUSSION

### Cluster Significance

As explained in the Materials and Methods section, to test for cluster significance of a given clustering algorithm, we apply the scoring functions defined in "Community scoring functions" to the clustering produced on the original graph and on randomized versions obtained by the method described in "Randomized graph". It should be expected that whenever the communities found by an algorithm on the original graph are significant, they will receive better scores than those found by the same algorithm on a graph with no actual community structure.

The results of computing these scores on the clustering obtained by the algorithms on each of the networks can be seen on Tables S1.1, S1.2, S1.3, S1.4, S1.5 and S1.6 (recall that ↑ identifies scores for which higher is best, and ↓ means lower is best). For each combination of scoring function and algorithm, we represent its value on the original network, its mean across multiple samples of its randomized switching model, and the percentile rank of the original score in the distribution of randomized graph scores. This percentile rank value serves as a statistical test of significance for each of the scores: a score is significant if its value is more extreme (either higher or lower, depending on its type) than most of the distribution.

It is important to note that some of the scores greatly depend on the number of clusters, and cannot adequately compare partitions in which that number differs. For instance, internal density can easily be high on small communities, while it will generally take lower values on bigger ones, even when they are very well connected. This can result in networks with no apparent community structure having high overall internal density scores just because they are partitioned into many small clusters.

In comparison, scores that combine both internal and external connectivity (conductance, normalized cut, out degree fractions), clustering coefficient, and modularity suffer less from this effect and seem more adequate in most circumstances. These also happen to be the scores that are invariant under the multiplication of the weights by positive constants (see 'Definitions').

We suggest focusing on the relative scores (the score of the actual network over the mean of the randomized ones) to simplify the process of interpreting the results, especially when trying to compare graphs of different nature. With relative scores, anything that differs significantly from 1 will suggest that the clustering is strong. For instance, in Fig. 3 we have the modularity of the stochastic block model for each algorithm, and for different values of the parameter λ (which will give increasingly stronger clusters). While the algorithms find results closer to the ground truth the bigger λ is, only the relative scores give us that insight. However, when comparing several clustering methods on the same network (and not

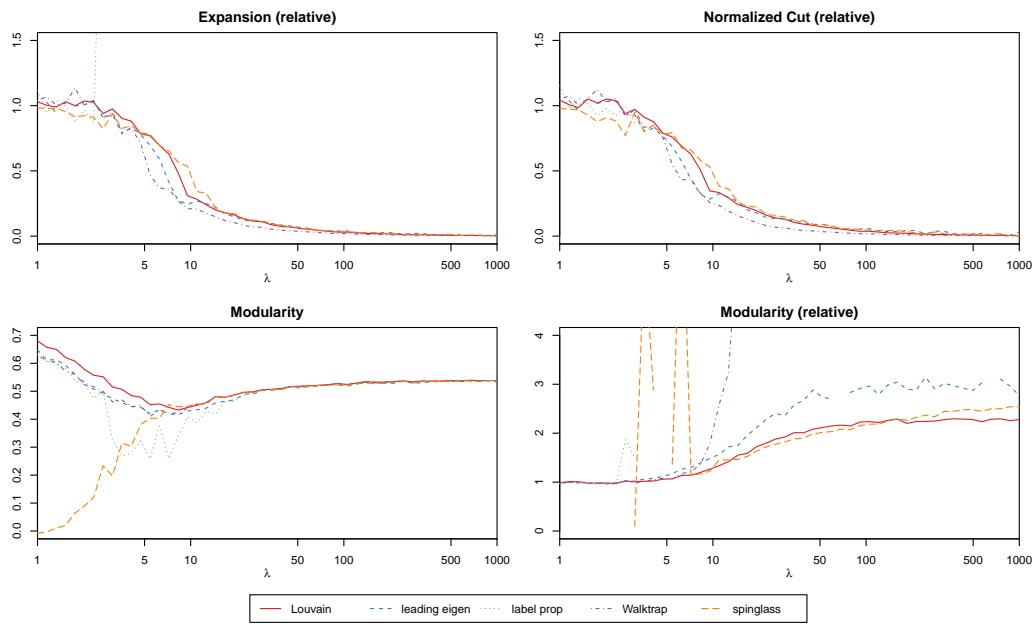

**Figure 3** **Scores of the weighted stochastic block model as a function of the parameter λ, for each of the algorithms.**

simply trying to determine if a single given method produces significant results), absolute scores are more meaningful to determine which one is best.

For the weighted stochastic block model graph, the clustering algorithms get results closer to the ground truth clustering the bigger the λ parameter is, as one would expect, and for λ > 30 the results perfectly match the ground truth clustering outcome in almost all cases (a bit earlier for the Louvain, Walktrap and spinglass cases, see Fig. 4). The relative scores match these results, and get better as λ increases as well (Fig. 3). Note that in Fig. 3, there are some jumps for the relative modularity in the spin-glass case, which are caused by the instability of this algorithm (see "Cluster stability"). This effect is no longer present when the structure of the network is stronger (λ > 8).

In Table S1.1, corresponding to λ = 15, we can see how for the Louvain algorithm, the scores are more extreme (lower when lower is better, higher when higher is better) than those of the randomized network in almost all circumstances. In the case of the leading eigenvector algorithm the scores are slightly worse, but almost all of them still fall within statistical significance (if we consider $p$-values < 0.05). In both cases, the only metric that is better in the random network is internal density, due to the smaller size of the detected clusters (which is why by itself internal density is not a reliable metric, as even in a network with very poor community structure it will be high for certain partitions into very small clusters that arise by chance). For both the label propagation and the Walktrap algorithms, the real network scores are not as close to the edge of the distribution of random scores, but they are still much better than the mean in all meaningful cases (the only exceptions

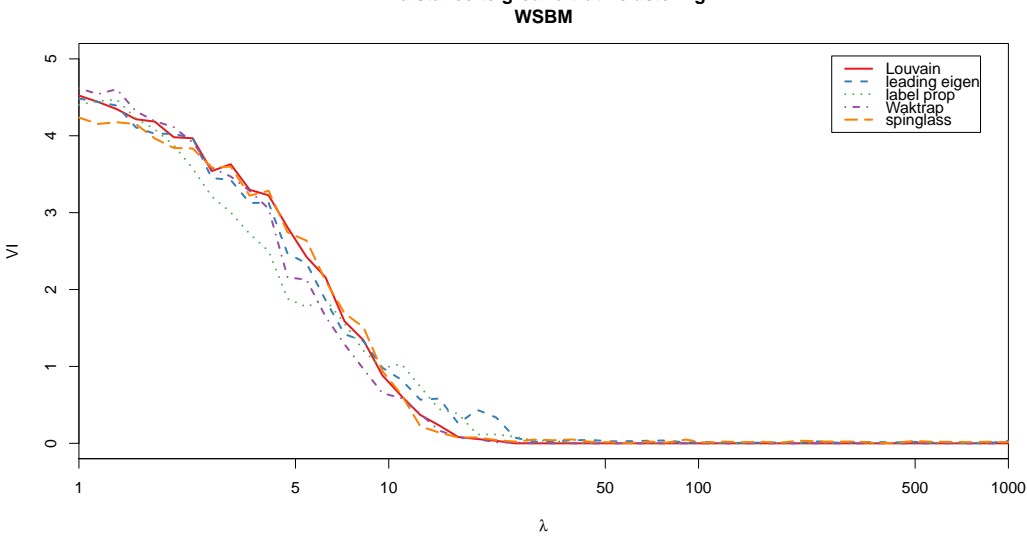

**Figure 4** **VI distance between the ground truth clustering and the result of each of the algorithms for the weighted stochastic block model (WSBM), as a function of the parameter λ.**

are the internal density and edges inside, which are hugely dependent on cluster size and are therefore inadequate to compare partitions with a different number of clusters).

In the case of the karate club network (Table S1.2), the label propagation algorithm gets the closest results to the ground truth clustering, and this is reflected in most scores being better than those of other methods. This does not apply to the modularity though, which is always higher for the Louvain and spin-glass, which produce identical clusters (this is to be expected, because Louvain is a method based on modularity optimization).

On the Forex graph (Table S1.3), we can see that both the leading eigenvector and Walktrap algorithms produce almost identical results splitting the network into two clusters, while the spin-glass algorithm splits it into three and Louvain into four. The scores which are based on external connectivity give better results to the Walktrap and leading eigenvector, while the spin-glass partition has slightly better clustering coefficient and better modularity (with Louvain having very similar values in those two scores).

It is also important to disregard the results of the scoring functions whenever the algorithms fail to distinguish any communities and either groups the whole network together or separates each element into its own cluster (such as the label propagation algorithm on the Forex network, seen in Table S1.3). In this case, the scores which are based on external connectivity will be optimum, as the cut $c_s$ of the partition is 0, but that of course does not give any information at all. In addition, the normalized cut and conductance could be not well defined in this case, as it is possible to have a division by 0 for some of the clusters.

As for the news on corporations graph (Table S1.4), the results and in particular the number of clusters vary greatly between algorithms (from 82 clusters for the Walktrap to only 2 for the label propagation). While the label propagation algorithm scores well on

some measures due to successfully splitting to very weakly connected components of the network, others such as the clustering coefficient or internal density are very low. Louvain and spin-glass have very similar scores across most measures and seem to be the best, though leading eigenvector does have better conductance and normalized cut. In this case the high variation in number of clusters across algorithms that still score highly could suggest that there is not a single predominant community structure in the network.

In the Enron graph (Table S1.5) Louvain also produces the best results for most scores, particularly in conductance and normalized cut, and it significantly surpasses all other algorithms while having larger clusters, with the only exception of label propagation, which partitions the network into much smaller clusters. The spin-glass algorithm stands out as having by far the worse results across all scores, even though its number of clusters (12) is the same as in leading eigenvector and similar to Louvain.

For the social network (Table S1.6), the Louvain, leading eigenvector and label propagation algorithms produce the same number of clusters (with spin-glass being also very close), which allows an unbiased comparison of scores. In this case, leading eigenvector has better results for almost all scores, except for clustering coefficient and modularity, for which Louvain is again the best algorithm. This huge disparity may be explained by the fact that modularity compares edge weights to a null model that considers the degrees of their incident vertices, and does not only discriminate between internal and external edges (as most of the scoring functions do).

Overall the Louvain algorithm seems to be the best at finding significant clusters, performing consistently well on a variety of weighted networks of very different nature. It is worth noting though that there are some limitations to it (and all modularity based methods in general) in terms of resolution limit (*Fortunato & Barthélemy, 2007*) that can appear when there are small communities in large networks, though there are methods to address it, such as the use of a resolution parameter (*Arenas, Fernández & Gómez, 2008*).

## Cluster stability

Using the non-parametric bootstrap method described in "Bootstrap with perturbation", we resample the networks 999 times ($R = 999$), apply clustering algorithms to them, and compare them to their original clustering with the metrics from "Clustering similarity measures". Stable clusterings are expected to persist through the process, giving small mean values of the variation of information, and high (close to 1) values of the normalized reduced mutual information and the Rand index. The results of the same method applied to the randomized versions of each network (see "Randomized graph") are also included, to have reference values for the stability of networks where there is no community structure. If the values of the clustering similarity measures, for the original and randomized networks, happened to be close together, that would suggest that the chosen algorithm produces a very unstable clustering on the network.

We observe in Table S1.7 that for the stochastic block model example graph, all algorithms except for spin-glass produce very stable clusters, which is consistent with the fact that we chose parameters to give it a very strong community structure. Meanwhile, clustering algorithms applied to the Zachary and Forex networks (Table S1.8 and S1.9)

produce clusters which are not as stable, but still much better than their baseline randomized counterparts. Note that the stability values for the label propagation algorithm in the Forex network (Table S1.9) should be ignored, as in that instance the output is a single cluster (see Table S1.3) which does not give any information. It is clear that while it works on less dense networks, the label propagation algorithm is not useful for complete weighted networks and it fails to give results that are at all meaningful.

On the news on corporations graph (Table S1.10) spin-glass is again the most unstable algorithm, with results for the RMI and ARI (which are both close to 0) that suggest that the clusters of the original network and all the resampled ones are completely unrelated. In this case the label propagation algorithm is the most stable, while the rest of the algorithms are not as good. This might be in part explained by the fact that its clusters are much bigger than in other networks, which allows them to remain strongly connected after small perturbations.

Finally, we observe that algorithms on the Enron graph (Table S1.11) produce the most unstable clusters out of all that were tested, which would suggest that the network does not have a single prevalent clustering structure that can be consistently detected, at least in the weighed graph configuration that we tested.

As a general remark on stability observed from all resulting experiments is that the spin-glass algorithm is the most unstable across the networks we tested, which are a diverse representation of different kinds of weighted networks.

## CONCLUSIONS

We have successfully observed how the community scoring functions, combined with the switching model, can easily help distinguish networks which have a community structure from others that do not. A combination of a network and a clustering algorithm can be said to produce significant clusters when their scores stand out from the distribution of scores produced by the same algorithm on the collection of randomized graphs produced by the switching model. The experiments conducted on the stochastic block model networks of varying community strength support this hypothesis. This will be useful when working with networks for which there is little information available, and one wants to determine whether the results obtained from any given clustering algorithms do reflect an actual community structure or if they are simply given by chance.

We recommend avoiding the scoring functions that can be heavily influenced by variables like the number of clusters or their size (like internal density, which favours smaller clusters), because the information they provide is hard to interpret in a systematic manner. In comparison, functions that combine internal and external connectivity, like conductance or normalized cut, seem more robust. However, we observed a tendency of these measures to favour partitions into fewer bigger clusters, which makes comparisons difficult when we want to compare partitions with a different number of parts. In contrast, both modularity and clustering coefficient do not seem to be so dependant on the number of communities in the partition, which is a relevant advantage.

We remark that our approach consists of a global analysis of the partitions, but it is possible to perform similar evaluations based on individual scores of each cluster. In this

case, some of the scoring functions that have not proved very useful might provide a more meaningful insight into the local structure of the partition, as it is possible to have both strong and weak clusters in the same network.

Additionally, the use of the switching model to generate randomized graphs provides a valuable point of reference, especially when we do not have much information on the structure of the network. The methods proposed here to test cluster significance can also be used with any other scoring functions, which could even be customized depending on the characteristics that one might want to prioritize in any given network clustering, such as giving more emphasis to internal connectivity, or external connectivity, or scores that naturally favour larger or smaller clusters.

In a more particular note, and according to our analyses, the Louvain algoritm, and to a lesser extent, the Walktrap algorithm, seem to be the most stable while producing significant clusterings, as specified by our scoring functions and across all networks considered. This reaffirms Louvain as one of the state-of-the-art clustering algorithms.

## APPENDIX

### Computational complexity of scoring functions
#### *Weighted clustering coefficient*

Let $\Gamma$ be the number of connected triplets in the graph and $\gamma$ the number of closed triplets (i.e., 3 times the number of triangles). As before $\Gamma(t)$ and $\gamma(t)$ are their respective values when only edges with weight greater or equal than $t$ are considered. Then, the clustering coefficient or transitivity is defined as:

$$\tilde{C} = \frac{1}{\bar{w}} \int_{t \geq 0} \frac{\gamma(t)}{\Gamma(t)} \, dt, \tag{23}$$

This is an integral of a step function that takes a finite number of values (bounded by the number of different edge weights) which we will compute as follows:

1. Construct a hash table of all edges with their corresponding weights to be able to search if there is an edge between any two edges (and obtain it's weight) in constant time. Complexity: $\mathcal{O}(m)$

2. Construct a hash table for each vertex containing all its neighbors. Can be done by iterating once over the edges and updating the corresponding tables at each step. This will be used to iterate over the connected triplets incident to each vertex. Complexity: $\mathcal{O}(m)$

3. Construct a sorted list containing the edge weights at which either a connected triplet or a triangle appears (i.e., the maximum edge weight of that triangle or triplet), and an associated variable for each indicating whether it corresponds to a triangle or a triplet. For this, we iterate over the connected triplets using the hash tables from step 2, and for each, we check if it also forms a triangle by checking the hash table from step 1 (which allows each iteration to be done in constant amortized time). This step has complexity $\mathcal{O}(\Gamma \log \Gamma)$, as the list has $\Gamma + \gamma$ elements, and $(\Gamma + \gamma) \in \mathcal{O}(\Gamma)$.

4. We iterate the list from step 3 and compute the cumulative sums of connected triplets and closed connected triplets (which correspond to $\gamma(t)$ and $\Gamma(t)$ for increasing values

of $t$ in the list). This gives us all values of $\frac{\gamma(t)}{\Gamma(t)}$, from which we compute the integral (Eq. (23)). This involves $\mathcal{O}(\Gamma)$ steps of constant complexity.

Therefore, the overall complexity of the algorithm is $\mathcal{O}(m + \Gamma \log \Gamma)$. Because $\Gamma$ is bounded by $m^2$, we can also express the complexity only in terms of $m$ (which will then be $\mathcal{O}(m^2)$), but that bound is not tight in most graphs.

### *Other scoring functions*

Computing $\tilde{m}$, $\tilde{m}_S$, $\tilde{c}_S$ (for all values of $S$), as well as all vertex degrees and out degrees has complexity $\mathcal{O}(m)$, as it can be done sequentially by reading the edge list and updating the appropriate values as necessary. This means that all scoring functions except for the clustering coefficient, which are derived from these values, can be computed very efficiently.

### Methods for counting contingency tables

To compute the number of contingency tables with fixed margins needed to obtain the value of the reduced mutual information, we mainly use the analytical approximation suggested by *Newman, Cantwell & Young (2020)*, which works whenever the number of clusters is substantially smaller than the number of nodes. This works well in most of the cases we study, except for the News graph when clustered with the Walktrap algorithm, which produces many single node clusters. For this case, we use a hybrid approach combining the analytical approximation for the clusters with more than one element, and then extending it to the full contingency table with the Markov chain Monte Carlo method described by *Diaconis & Gangolli (1995)*. This estimates the size of the set by defining a nested sequence of subsets and obtaining the ratio between the size of each one of them and its predecessor with a Monte Carlo approximation.

Our solution consists of sorting and rearranging the rows and columns of the original contingency table so that smaller elements sit at the top left part of the table. Then, we use the analytical approximation on the submatrix formed by rows and columns with sums strictly greater than one (which will sit on the bottom right corner). This will be the size of the first subset of the chain, and the rest are estimated successively with the Markov chain Monte Carlo method.

This method works well on the contingency tables generated by the Walktrap clustering on our News graph, unlike the analytical approximation alone, which is inaccurate, or the Monte Carlo method alone, which is much slower. However if the RMI is to be used to compare partitions of very large graphs, establishing some general criteria to determine the largest subset that can be analytically estimated with enough accuracy might be needed, with the goal of minimizing the need for costly Monte Carlo approximations. This topic has a lot of potential to be studied in future work, and which we hope to address in the future in our *clustAnalytics* package.

### Funding

This work was supported by grant TIN2017-89244-R from MINECO (Ministerio de Economía, Industria y Competitividad) and the recognition 2017SGR-856 (MACDA) from AGAUR (Generalitat de Catalunya). The funders had no role in study design, data collection and analysis, decision to publish, or preparation of the manuscript.

### Grant Disclosures

The following grant information was disclosed by the authors:
MINECO (Ministerio de Economía, Industria y Competitividad): TIN2017-89244-R.
AGAUR (Generalitat de Catalunya): 2017SGR-856 (MACDA).

### Competing Interests

The authors declare there are no competing interests.

### Author Contributions

- Argimiro Arratia conceived and designed the experiments, performed the experiments, analyzed the data, prepared figures and/or tables, authored or reviewed drafts of the paper, and approved the final draft.
- Martí Renedo Mirambell conceived and designed the experiments, performed the experiments, analyzed the data, performed the computation work, prepared figures and/or tables, authored or reviewed drafts of the paper, and approved the final draft.

### Data Availability

The data and code files are available in the Supplemental File and at GitHub: https://github.com/martirm/cluster_assessment. The raw data is inside the data folder.

### Supplemental Information

Supplemental information for this article can be found online at http://dx.doi.org/10.7717/peerj-cs.600#supplemental-information.

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
