# Peer review of "Clustering assessment in weighted networks"

_PeerJ Computer Science, doi:10.7717/peerj-cs.600_

## Round 0.1 · original submission · Major Revisions

Please give a point-to-point response to the reviewers and also highlight all changed places in the revised paper.

·

Basic reporting

"Clear and unambiguous, professional English used throughout."
Yes.

"Literature references, sufficient field background/context provided."
Some definitions, although they appear to be right, have not been referenced.

"Professional article structure, figures, tables. Raw data shared."
Yes.

"Self-contained with relevant results to hypotheses."
Yes.

"Formal results should include clear definitions of all terms and theorems, and detailed proofs."
In equation 1 explain the steps that were performed.

Experimental design

"Original primary research within Aims and Scope of the journal.
Research question well defined, relevant & meaningful. It is stated how research fills an identified knowledge gap."
Yes.

"Rigorous investigation performed to a high technical & ethical standard."
Yes.

"Methods described with sufficient detail & information to replicate."
The Louvain algorithm has parameters that can be modified, the parameters used must be presented.
The 1/2 used in equation 21 can be applied to graphs in addition to the directed ones, for example in weighted graphs (Ex: value -0.5/0.5).

Validity of the findings

"Impact and novelty not assessed. Negative/inconclusive results accepted. Meaningful replication encouraged where rationale & benefit to literature is clearly stated."
The article presents developments for the area.

"All underlying data have been provided; they are robust, statistically sound, & controlled."
The data in section 2.7 must be made available for download or the work must explain where they are.

"Conclusions are well stated, linked to original research question & limited to supporting results."
The article could feature a discussion session

"Speculation is welcome, but should be identified as such."
No comment.

Additional comments

The article brings significant contributions to the area, but some topics must be changed to better understand the reader.
The "scoring functions" of section 2.1 must be referenced.
Present equation 2 also as a summation.
In equation 18 add parentheses.
The computational complexity of the methods used must be explained. Many use the adjacency matrix, it has a high cost of n^2.
Say the scale of the graphics. The y-axis in figure 4 did not appear to me.

Reviewer 2 ·

Basic reporting

no comment

Experimental design

no comment

Validity of the findings

no comment

Additional comments

The authors proposed a systematic approach to assess the clustering results in weighted networks. The theoretical presentations are sufficient and the experiments can support the evaluation metrics. However, some problems should be addressed as follows:
1. For the abstract, the authors should give some specific performance analyses for the verification experiments.
2. For the introduction, the background and the research significance should be re-expressed. Furthermore, the challenges and contributions should be explicitly summarized. There needs a paragraph at the end of this part to describe the organization of this paper.
3. There exist a lot of acronyms without definitions for the first time, such as VAR, SBM, WSBM, and wSBM, etc., and some full names have been referred to repeatedly after the first acronyms, such as reduced mutual information. Please check the whole manuscript to revise them.
4. In section 2.1, the definition and presentation of modularity should be given sub-title to highlight liking other metrics. Further, the variable Q has the same name with different definitions as in Section “Number of iterations”. Please revise the second variable name. In a complex network, Q is always considered modularity.
5. For Figure 2, the “Forex club graph” should be corrected as “Forex graph” and “club” is “karate club graph”. For the second sub-figure, are the lines of variance and VI overlapped, please give the related explanation.
6. Based on Figures 4 and 5, the trend analysis referred to the lamda>20 and lamda=15 may be inaccurate. Furthermore, why Figure 4 has not compared with the algorithm spin-class?
7. In Section 3, some of the analysis referred to the Figures or Tables are inaccurate, for example, “the leading eigenvector algorithm the effect isn’t as pronounced” in line 451 on page 13, “internal density” in line 476 on page 13. Please check the related content in terms of the tables.
8. Please improve the pixel for each figure.
9. The manuscript has been presented in a disorganized way, the theoretical presentation and the experimental design should be given in different sections. Furthermore, the experiments and analysis should be reorganized.

Reviewer 3 ·

Basic reporting

At a glance, in this paper, the authors present metrics to evaluate clustering algorithms, focused on weighted graphs.
Known the performance of community detection and clustering methods is important, especially when one does not know a priori the network topology or properties.
Despite the importance of evaluating the performance of such methods, the current work, in fact, is not able of telling readers how well a method performs. As the authors themselves say at the conclusion, the metrics help distinguish networks, which have a community structure from others that do not. It does not tell how well the metric can cluster or form groups, maintaining properties.

Experimental design

A second issue of the paper is related to its organization. Authors mix everything in a single section (materials and methods). The way they present it, it is not clear (i) what they are proposing. (ii) how they are planning to evaluate their proposal.
I suggest authors work on article organization, first by presenting the things you propose.
Second, in the following section, I suggest presenting the experimental design. In this section, you can present carefully all the material and methods.
You detail the way you generate random graphs. The way you generate perturbations.
Moreover, how do you design your evaluations (and comparisons)?
At this point, I think you have 3 steps to evaluate the metrics you are proposing:
The design of comparison with ground truth (the random graphs).
The design of the evaluations of synthetic graphs
In addition, the evaluation of real graphs.

In this same section, you may overview the clustering methods you will use in your evaluations.


Third, and finally, you present the evaluation section. In this section, you present your findings.
You must show how your metrics evidence that clustering methods are performing well or not.
In special, I suggest comparing to the state of the art. As your claim is “metrics to unweighted graphs are not appropriate… then you are proposing new metrics”. You must show this, in your result section. (that metrics for unweighted graphs are not appropriate AND… that your metrics outperform somehow existing state of the art).

Validity of the findings

In sum, the findings do no support the main goal of the paper. As you are presenting new metrics because the existing metrics do not perform well (or are not appropriate) to weighted graphs, you must show this clearly. Moreover, it is not clear that your metrics show how a clustering method performs. The way you conclude the article, you are just telling us that you can differentiate random graphs from graphs with clusters.

Additional comments

The abstract is confusing. I suggest focusing on the real paper contribution.
Developing a way to generate random graphs is not a contribution of the article. There are many methods to this purpose.

---

## Round 0.2 · Minor Revisions

One reviewer still have some minor comments that need to be addressed before further consideration of the paper.

·

Basic reporting

All of my suggestions have been correctly modified.
In my opinion, the article is suitable for submission.

Experimental design

...

Validity of the findings

...

Additional comments

...

Reviewer 2 ·

Basic reporting

no comment

Experimental design

no comment

Validity of the findings

no comment

Additional comments

The authors have revised the manuscript according to the comments in detail. However, the figures should be re-generated with high quality, and resolution, font size, etc. Especially, the figure 1 and figure 2 are poor quality.

Reviewer 3 ·

Basic reporting

My biggest criticism about the article remains about the lack of evaluation of how well a method performs. However, the authors clearly stated and answered me their main objectives.
I notice the improvements on the organization of the article and I have also followed the other reviewers questions and the authors questions.
I´m not an English language specialist. However, I think authors could use more formal of negative tenses in most part of the text, instead of abbreviation.
Minor suggestion: in line 52, page 6 – “that measure some topological characteristics.” -> can you specify the topological characteristics here to be more precise?
Figure 2 – labels are messy. One on top of other.

Experimental design

no comment

Validity of the findings

no comment

Additional comments

no comment

---

## Round 0.3 · accepted · Accept

The revised paper has addressed the reviewers' questions and can be accepted.